

# Effective attributed network embedding with information behavior extraction

Ganglin Hu[1], Jun Pang[2] and Xian Mo[3]

[1] College of Computer & Information Science, Centre for Research and Innovation in Software Engineering, Southwest University, Chongqing, Chongqing, China

[2] Faculty of Science, Technology and Medicine & Interdisciplinary Centre for Security, Reliability and Trust, University of Luxembourg, Luxembourg, Luxembourg

[3] School of Information Engineering, Ningxia University, Yinchuan, Ningxia, China

## ABSTRACT

Network embedding has shown its effectiveness in many tasks, such as link prediction, node classification, and community detection. Most attributed network embedding methods consider topological features and attribute features to obtain a node embedding but ignore its implicit information behavior features, including information inquiry, interaction, and sharing. These can potentially lead to ineffective performance for downstream applications. In this article, we propose a novel network embedding framework, named information behavior extraction (IBE), that incorporates nodes' topological features, attribute features, and information behavior features within a joint embedding framework. To design IBE, we use an existing embedding method (*e.g.*, SDNE, CANE, or CENE) to extract a node's topological features and attribute features into a basic vector. Then, we propose a topic-sensitive network embedding (TNE) model to extract a node's information behavior features and eventually generate information behavior feature vectors. In our TNE model, we design an importance score rating algorithm (ISR), which considers both effects of the topic-based community of a node and its interaction with adjacent nodes to capture the node's information behavior features. Eventually, we concatenate a node's information behavior feature vector with its basic vector to get its ultimate joint embedding vector. Extensive experiments demonstrate that our method achieves significant and consistent improvements compared to several state-of-the-art embedding methods on link prediction.

# INTRODUCTION

Network embedding (NE) aiming to map nodes of networks into a low-dimensional vector space, has been proved extremely useful in many applications, such as node classification (*Perozzi, Al-Rfou & Skiena, 2014*; *Tang et al., 2015*), node clustering (*Cao, Lu & Xu, 2015*), link prediction (*Grover & Leskovec, 2016*). A number of network embedding models have been proposed to learn low-dimensional vectors for nodes *via* leveraging their structure and attribute information in the network. For example, spectral clustering is an early method for learning node embedding, including models such as DGE (*Perrault-Joncas & Meila, 2011*), LE (*Wang, 2012*) and LLE (*Roweis & Saul, 2000*). Matrix

Corresponding author
Ganglin Hu,
huganglin88@outlook.com

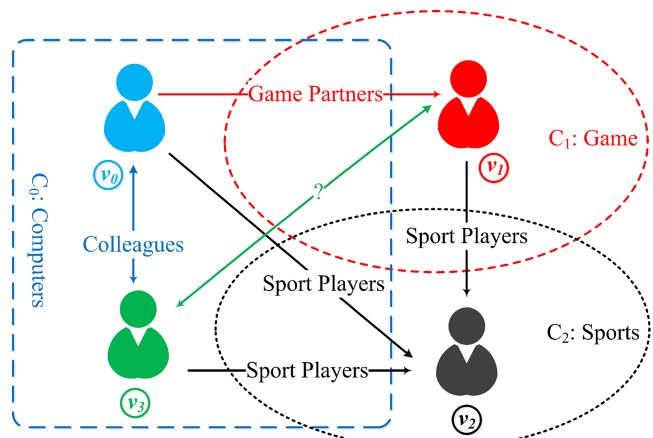

**Figure 1 Node information behavior of multiple topic-based communities** ($C_0$, $C_1$, and $C_2$ are topic-based communities, and $v_0$, $v_1$, $v_2$, $v_3$ are nodes, and in the meantime $(v_0, v_3) \in C_0, v_1 \in C_1$, $v_2 \in C_2$). Nodes interact in intra-community, such as nodes $v_0$ and $v_3$. Nodes in different communities may interact with each other, such as nodes $v_0$ and $v_1$, $v_1$ and $v_2$, $v_2$ and $v_3$. Due to the existence of bridge nodes $v_0$ or $v_2$, nodes $v_1$ and $v_3$ may have a link which is not represented in the current network.

decomposition is another important method for learning node embedding, for example, GraRep (*Cao, Lu & Xu, 2015*) and TADW (*Yang et al., 2015*). Spectral clustering and matrix decomposition embedding methods usually have high computational complexity with the increasing network size. In recent years, various network embedding methods have been proposed using random walk-based methods, which are faster and more effective, including DeepWalk (*Perozzi, Al-Rfou & Skiena, 2014*), Node2Vec (*Grover & Leskovec, 2016*), LINE (*Tang et al., 2015*), and SDNE (*Wang, Cui & Zhu, 2016*). More recently, deep learning and attention mechanisms are used to generate network embeddings, including GCN (*Kipf & Welling, 2016*), GAT (*Veličković et al., 2018*) and CANE (*Tu et al., 2017*), which extract structure, text, topic, and other heterogeneous feature information more effectively. In addition to the above methods, a few role-based approaches of network embedding (*Jiao et al., 2021*) have been proposed recently, where a role-based network embedding (*Zhang et al., 2021*) captures the structural similarity and the role information. Nevertheless, these methods are all limited with focusing on the network topological structure and attributed information while ignoring the implicit relationship between information. In reality, information has interactive behavior in some real-world networks such as social networks and citation networks, where the nodes have information behavior (*Pettigrew, Fidel & Bruce, 2001*), including information inquiry, information access, and information sharing. Therefore, we introduce the concept of informational behavior to deal with the information interaction. In real-world social networks and citation networks, it is intuitive that all of the nodes naturally prefer to interact with similar nodes. In this way, the exchange and sharing of information between nodes are more efficient, which is also the reason for the formation of various topic networks communities. For example, due to different majors, three topic-based communities, $C_0$, $C_1$, and $C_2$, have been formed (as illustrated in Fig. 1). In these

communities, nodes interact within both intra-community (such as nodes $v_0$ and $v_3$) and inter-community (such as nodes $v_0$ and $v_1$, $v_1$ and $v_2$, $v_2$ and $v_3$). This means that one node may communicate and share information in various topics when interacting with neighboring nodes of different communities and build bridges among nodes that are not directly connected, such as nodes $v_1$ and $v_3$ may have a link because nodes $v_0$ or $v_2$ acts as a bridge, but this link is not observed. It can be seen that these information behaviors are very important features, and the representation vectors for nodes without information behavior features are incomplete. However, these existing embedding methods are not able to cope with the information behavior of nodes.

To tackle the above-identified problems, we make the following contributions: (1) We demonstrate the importance of integrating structure features, attributed features, and node's information behavior features in attribute networks. (2) We propose a joint embedding framework IBE to add the information behavior feature vector to a basic vector to obtain a final joint embedding vector, which has never been considered in the literature. The basic vector is generated by one of existing embedding methods. Within the framework, we design an algorithm ISR to generate a topic-sensitive vector for a given topic, and then we get information behavior feature vectors by matrix transposing a topic-sensitive embedding matrix composed of all topic-sensitive vectors. (3) We conduct extensive experiments in real-world information networks. Experimental results prove the effectiveness and efficiency of the proposed ISR algorithm and IBE framework.

The rest of the article is organized as follows. "Related Work" discusses several related works. We provide some definitions and problem formulation in "Problem Definition". "Our Approach" presents in detail our proposed IBE framework and ISR algorithm. We then show experimental results in "Experiments" before concluding the article in "Conclusion and Future Work".

## RELATED WORK

In the last few years, a large number of NE models have been proposed to learn node network embedding efficiently. These methods can be classified into two categories based on structural information and attributes: (1) SNE methods by considering purely structural information; and (2) ANE methods by considering both structural information and attributes. In this section, we briefly review related work in these two categories.

### SNE methods

DeepWalk (*Perozzi, Al-Rfou & Skiena, 2014*) employs Skip-Gram (*Mikolov et al., 2013*) to learn the representations of nodes in the network. It uses a random selection of nodes and truncated random walk to generate random walk sequences of fixed length. Subsequently, these sequences are transported to the Skip-Gram model to learn the distributed node representations. LINE (*Tang et al., 2015*) studies the problem of embedding very large information networks into low-dimensional vector spaces. Node2vec (*Grover & Leskovec, 2016*) improves the strategy of random walk and achieves a balance between BFS and DFS. SDNE (*Wang, Cui & Zhu, 2016*) proposes a semi-supervised deep model, which can learn a highly nonlinear network structure. It combines

the advantages of first-order and second-order estimation to represent the global and local structure attributes of the network. Besides, there are many other SNE methods (*Goyal & Ferrara, 2017*), which systematic analysis of various structural graph embedding models, and explain their differences. Nevertheless, these methods fully utilize structural information but do not consider attribute information.

## ANE methods

CANE (*Tu et al., 2017*) proposed an approach of network embedding considering both node text information of context-free and context-aware. CENE (context-enhanced network embedding) (*Sun et al., 2016*) regards text content as a special kind of nodes and leverages both structural and textural information to learn network embeddings. TopicVec (*Li et al., 2016*) proposes to combine the word embedding pattern and document topic model. JMTS (*Alam, Ryu & Lee, 2016*) proposes a domain-independent topic sentiment model to integrate topic semantic information into embedding. ASNE (*Liao et al., 2018*) adopts a deep neural network framework to model the complex interrelations between structural information and attributes. It learns node representations from social network data by leveraging both structural and attribute information. ABRW (*Hou, He & Tang, 2018*) reconstructs a unified denser network by fusing structural information and attributes for information enhancement. It employs weighted random walks based network embedding method for learning node embedding and addresses the challenges of embedding incomplete attributed networks. AM-GCN (*Wang et al., 2020*) is able to fuse topological structures, and node features adaptively. Moreover, there exist quite a few survey papers (*Jiao et al., 2021*; *Zhang et al., 2020*; *Daokun et al., 2017*; *Peng et al., 2017*; *Cui et al., 2019*), which provide a comprehensive, up-to-date review of the state-of-the-art network representation learning techniques. They cover not only early work on preserving network structure, but also a new surge of incorporating node content and node labels.

It is challenging to get network embedding considering attributes of local context and topic due to its complexity. Quite a few works have been carried out on this issue. However none of them consider node information behavior features in attributed networks.

## PROBLEM DEFINITION

In this section, we present the necessary definitions and formulate the problem of link prediction in attributed networks.

**Definition 1 (Networks)** *A network can be represented graphically: $G = (V, E, \Delta, A)$, where $V = \{v_0, v_1, \ldots, v_{(|V|-1)}\}$ represents the set of nodes, and $|V|$ is the total number of nodes in G. $E \subseteq V \times V$ is the set of edges between the nodes. $\Delta = \{\delta_0, \delta_1, \ldots, \delta_{(\tau-1)}\}$ is a set, where $\delta$ represents a topic and it is also a topic-based node label identified the topic of the node, $\tau$ is the total number of topics. A is a function which associates each node in the network with a set of attributes, denoted as A(v).*

**Definition 2 (Adjacent-Node and Node degree)** *An adjacent-node set of node $v \in V$ is defined as $N_v = \{v' : (v, v') \in E\}$. $v^{degree}$ is the number of nodes in the adjacent-node set of v, called the degree of node v.*

**Definition 3 (Topic-based community)** *Each node v has topic-based labels to identify the topics it belongs to. A topic-based community is a node-set that consists of the nodes with same topic-based label. Here, we define a topic-based community as $C_\delta = \{v : v \in V, \delta \in \Delta\}$, and the node number in $C_\delta$ is defined as $|C_\delta|$. The topic-based community set is represented as $C^\Delta = \{C_{\delta_0}, C_{\delta_1}, \ldots, C_{\delta_{(\tau-1)}}\}$ ($\bigcup_{j=0}^{(\tau-1)} C_{\delta_j} = V$), where $C^\Delta$ is a set of all topic-based communities. Note that we assume that each node has at least one topic-based label, and one node can belong to several topic-based communities.*

**Definition 4 (Information behavior)** *Information behavior is an individual node's action in a topic category, including information inquiry, information access, information interaction, information sharing, etc.*

A message passing is a process of global information recursive, such as GCN. Different from the mechanism of message passing, the information behavior is a process of local-independent information aggregation.

**Definition 5 (Importance Score)** *Given a topic $\delta$, the importance score $x_i$ ($0 \le i < |V|$) of node $v_i$ is computed as follows:*

$$x_i = \beta * m_i + (1 - \beta) * s_i \tag{1}$$

*where $0 \le \beta \le 1$ is a hyper-parameter, $m_i$ and $s_i$ are the adjacent score and community score of node $v_i$, respectively. $v_i$'s adjacent score $m_i$ is defined as the weighted importance score of its adjacent nodes:*

$$m_i = \sum_{v_k \in N_i} \frac{x_k}{\left(v_k^{degree}\right)},$$

*where $x_k$ is the importance score of $v_k$, $v_k^{degree}$ is the degree of node $v_k$, and $N_i$ is the adjacent-node set of node $v_i$. Moreover, $v_i$'s community score $s_i$ with respect to the topic $\delta$ is defined as:*

$$s_i = \begin{cases} \frac{1}{|C_\delta|}, & if \ v_i \in C_\delta \\ 0, & otherwise \end{cases}$$

*where $C_\delta$ is a topic-based community and $|C_\delta|$ is the number of nodes in $C_\delta$.*

The importance score $x_i$ of node $v_i$ reflects the interaction between $v_i$ and its adjacent nodes $N_i$, as well as the level of correlation between $v_i$ and its topic-based community $C_\delta$ ($\delta \in \Delta$).

**Definition 6 (Topic-sensitive vector)** *Given a topic $\delta$, the importance scores of all nodes can be used to form a topic-sensitive vector $\overrightarrow{\gamma^\delta} = (x_0^\delta, x_1^\delta, \ldots, x_{(|V|-1)}^\delta)$ ($\delta \in \Delta, \overrightarrow{\gamma^\delta} \in \mathbb{R}^{|V|}$).*

The learning process of the topic-sensitive vector is a repetitive iteration process for computing the importance scores of all nodes. The initial importance scores is

$x_i = \beta * \frac{1}{|V|} + (1 - \beta) * s_i$ ($\beta = 0.85$ is a hyper-parameter; $s_i = \frac{1}{|C_\delta|}$ if $v_i \in C_\delta$, otherwise $s_i = 0$.). Illustrated by Eq. (1), each iteration is an one-order aggregate operation of adjacent scores and community score and a new value $\frac{1}{|C_\delta|}$ is added to $x_i$ for node $v_i$. After a number of iterations (*i.e.*, higher-order aggregations), the ratio between $x_i$ of each node $v_i$ will

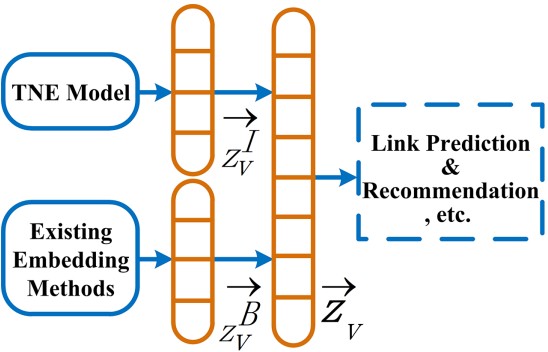

**Figure 2 Our information behavior extraction framework (IBE).**

stabilize, but the $x_i$ for each node $v_i$ will continue to grow due to the continuous addition of the $v_i$'s community score $s_i$. So, after each iteration, we normalise the $x_i$ for every node by

$$\hat{x}_i = \frac{x_i}{\sqrt{\sum_{i=0}^{n-1} (x_i)^2}}, \quad (0 \le i < |V|).$$

In this way, the $x_i$ obtained from this iteration process will eventually converge.

**Attributed network embedding.** Given an attributed network $G = (V, E, \Delta, A)$, our goal is to extract the node information behavior features and learn an information behavior feature vector $\vec{z_v^I} \in \mathbb{R}^d$ ($d = K * \tau, d \ll |V|$) for each node $v$. The distance between $\vec{z_v^I}$ and $\vec{z_{v'}^I}$ is the information behavior similarity of two nodes $v, v'$ ($v, v' \in V$). After that, node information behavior feature vector $\vec{z_v^I}$ is added to the node basic vector $\vec{z_v^B} \in \mathbb{R}^{d'}$ ($d' \ll |V|$) generated by one of existing embedding methods to get the ultimate joint embedding vector:

$$\vec{z_v} = ([\vec{z_v^I} \| \vec{z_v^B}]) \in \mathbb{R}^{d+d'} \tag{2}$$

where $[\cdot\|\cdot]$ denotes concatenating two vectors end to end. Nodes with similar network-structure features, node-attribute features, and information-behavior features are close to each other in the embedding space $\mathbb{R}^{d+d'}$.

## OUR APPROACH

In this section, we introduce our method of information behavior features extraction. We firstly propose our framework IBE (Information behavior extraction framework) which elaborates the components of node joint embedding vectors. Then, we present the model TNE (Topic-sensitive network embedding model) which describes the process of generating information behavior feature vectors.

### Information behavior extraction framework (IBE)

The information behavior features generated based on topics are complementary to features of existing models. As shown in Fig. 2, data sources of the framework IBE consist

**Figure 3 Overview of TNE model.**   

of two parts. One is the network embedding $Z^B$ generated by one of existing embedding methods, and the other is $Z^I$, where $Z^B \in \mathbb{R}^{|V| \times d'}$ and $Z^I \in \mathbb{R}^{|V| \times d}$ ($d'$, $d \ll |V|$) are embedding matrix consisting of the embedding vectors $\overrightarrow{z_v^B}$ and $\overrightarrow{z_v^I}$ of nodes $V$, respectively. We linearly concatenate the embedding matrix $Z^B$ and $Z^I$ to generate a joint embedding matrix $Z$, which can be used for link prediction, recommendation, and other tasks in attributed networks.

## Topic-sensitive network embedding model (TNE)

In this section, we present the process of extracting node information behavior features. As shown in Fig. 3, the TNE model consists of two parts: an ISR algorithm (Importance score rating algorithm) and a topic-sensitive embedding matrix ($\Gamma$) transposing.

### Importance score rating algorithm (ISR)

The ISR algorithm is used to get the importance scores of all nodes (illustrated by Eq. (1) and Definition 5) and generates a topic-sensitive vector under a given topic. We firstly input raw data including a node set $V$, an adjacent-node set $N_v$, and a topic-based community $C_\delta$ to ISR (see Algorithm 1) and then simulate the iteration process of node information behavior under the given topic $\delta$. When the importance scores of all nodes stabilize, the iteration is terminated. We propose *loss* as a metric for iteration termination, which is calculated as follows.

$$loss = \sum_{i=0}^{|V|-1} (|x_i - x_i'|) \tag{3}$$

where $x_i$ is the current iterative importance score for the node $v_i$ and $x_i'$ is the importance score of the previous iteration for the node $v_i$.

After the iteration is done, we can obtain a $|V|$-dimensional topic-sensitive vector $\overrightarrow{\gamma^\delta} = (x_0, x_1, \ldots, x_{(|V|-1)})$ (illustrated by Definition 6), consisting of importance scores of all nodes under a given topic $\delta \in \Delta$.

Given a topic $\delta$, the computing steps of topic-sensitive vector $\overrightarrow{\gamma^\delta}$ are given as follows (see Algorithm 1):

1. For a node $v_i$, the line 6–8 is used to compute its adjacent scores $m_i$ of all of neighbors $N_i$ and the line 9–13 are used to compute its community score $s_i$. Using $m_i$ and $s_i$, the importance scores $x_i$ of node $v_i$ is finally calculated out in the first statement of line 14.

2. The second layer loop (line 5, line 14–15) is used to assemble the importance scores of all nodes to generate a list $\gamma^\delta = [x_0, x_1, \ldots, x_{(|V|-1)}]$ in program which is the topic-sensitive vector $\overrightarrow{\gamma^\delta} = (x_0, x_1, \ldots, x_{(|V|-1)})$ ($\overrightarrow{\gamma^\delta} \in \mathbb{R}^{|V|}$).

---

**Algorithm 1 ISR algorithm**

---

**Input:** $\delta$: a topic, $\delta \in \Delta$;

  $V$: a node set of network $G$;

  $|V|$: node number of network $G$;

  $N_i$: a adjacent node set of node $v_i$;

  $v_i^{degree}$: degree of node $v_i \in V$;

  $C_\delta$: a topic-based community, $C_\delta \in C^\Delta$;

  $|C_\delta|$: node number of topic-based community $\delta$;

  $\beta = 0.85$: a hyper-parameter, imposing the ratio between $m_i$ and $s_i$;

**Output:** $\gamma^\delta$;

1  $\gamma^\delta = [x_0, x_1, ..., x_{(|V|-1)}] = [\frac{1}{|V|}, \frac{1}{|V|}, ..., \frac{1}{|V|}] = \tilde{\gamma}^\delta$, initializing every element of list $\gamma^\delta$ and $\tilde{\gamma}^\delta$ with $\frac{1}{|V|}$

  in a given topic $\delta \in \Delta$, where $\tilde{\gamma}^\delta$ is used to temporarily store a topic sensitive vector;

2  loss = 30;

3  **while** $loss > \frac{1}{|V|}$ **do**

4    loss = 0; norm = 0;

5    **for** *each* $v_i \in V$ **do**

6      **for** *each* $v_k \in N_i$ **do**

7        $x_k = \gamma^\delta[v_k]$; $m_i = m_i + \frac{x_k}{(v_k^{degree})}$;

8      **end**

9      **if** $v_i \in C_\delta$ **then**

10        $s_i = \frac{1}{|C_\delta|}$;

11      **else**

12        $s_i = 0$;

13      **end**

14      $x_i = \beta * m_i + (1 - \beta) * s_i$; $\gamma^\delta[v_i] = x_i$; norm = norm + $x_i^2$;

15    **end**

16    norm = $\sqrt{norm}$;

17    **for** *each* $v_i \in V$ **do**

18      $\gamma^\delta[v_i] = \frac{\gamma^\delta[v_i]}{norm}$; loss = loss + $(|\gamma^\delta[v_i] - \tilde{\gamma}^\delta[v_i]|)$; $\tilde{\gamma}^\delta[v_i]) = \gamma^\delta[v_i]$;

19    **end**

20 **end**

21 return $\gamma^\delta = [x_0, x_1, ..., x_{(|V|-1)}]$;

---

3. The third statement of line 14 and line 16 are use to calculate the Euclidean norm *norm*. The line 17–19 is used to normalise the importance scores of all nodes by the Euclidean norm (the first statement of line 18) and calculate the sum of *loss* for all node (the second statement of line 18). The third statement of line 18 is used to update the

value $\tilde{\gamma}^\delta[v_i]$ of node $v_i$, which is the importance score and will be used in the next iteration.

4. The first layer loop (line 3, line 20) is used to control the iterations.

In the ISR algorithm, an information behavior feature vector is generated without an increase in time complexity. We define $L$ as the number of iterations, $n$ represents the number of nodes in attributed networks, and $v^{degree}$ represents the degree of node $v$. The time complexity of the ISR algorithm to generate a topic-sensitive vector $\overrightarrow{\gamma^\delta} = (x_0, x_1, ..., x_{(|V|-1)})$ is $\mathcal{O}(L \cdot n \cdot v^{degree})$. Because $L \cdot v^{degree}$ and $n$ are of the same order of magnitude, the time complexity of the ISR algorithm is thus $\mathcal{O}(n^2)$.

*Topic-sensitive embedding matrix ($\Gamma$) transposing*

For $\Delta = \{\delta_0, \delta_1, ..., \delta_{(\tau-1)}\}$, the $\Gamma$ matrix transposing calls the Algorithm 1 to get every topic-sensitive vectors $\overrightarrow{\gamma^\delta}$ in each topic $\delta \in \Delta$. After all $\tau$ topic-sensitive vectors are obtained, we combine the $\tau$ topic-sensitive vectors to form a topic-sensitive embedding matrix $\Gamma = (\overrightarrow{\gamma^{\delta_0}}, \overrightarrow{\gamma^{\delta_1}}, ..., \overrightarrow{\gamma^{\delta_{(\tau-1)}}})^T$. Ultimately, $Z^I$ is obtained by the $\Gamma$ matrix transposing, as illustrated by Eq. (4).

$$Z^I = (\overrightarrow{z_{v_0}^I}, \overrightarrow{z_{v_1}^I}, ..., \overrightarrow{z_{v_{(|V|-1)}}^I})^T = \Gamma^T = (\overrightarrow{\gamma^{\delta_0}}, \overrightarrow{\gamma^{\delta_1}}, ..., \overrightarrow{\gamma^{\delta_{(\tau-1)}}}) \tag{4}$$

Each row of $Z^I$ is an information behavior feature vector $\overrightarrow{z_v^I}$ for node $v$ and the dimension of $\overrightarrow{z_v^I}$ is $d = \tau$.

## Generating node joint embedding vectors based on `IBE`

$Z^B$ is a basic embedding matrix trained by one of existing embedding methods. Each row of the $Z^B$ is a basic vector $\overrightarrow{z_v^B}$ for a node $v$ generated by one of existing embedding methods.

Before getting $Z$ by Eq. (7) according to the framework `IBE`, we firstly enlarge $Z^I$ or $Z^B$ by $\lambda$ (Eq. (5)) so that the element values of $\lambda * Z^I$ and $Z^B$ or $Z^I$ and $\frac{Z^B}{\lambda}$ are of the same order of magnitude, and the $\lambda$ is calculated as follows.

$$\lambda = \frac{\overline{|b|}}{\overline{x}} = \left( \frac{\sum_{i=0}^{|V|-1} \sum_{j=0}^{d-1} |b_{ij}|}{|V| * d} \right) \div \left( \frac{\sum_{i=0}^{|V|-1} \sum_{j=0}^{\tau-1} x_{ij}}{|V| * \tau} \right) \tag{5}$$

where $\overline{|b|}$ is the average of all elements in $Z^B$, and $\overline{x}$ is the average of all elements in $Z^I$. And then we enlarge the element values of $Z^I$ or $Z^B$ who has the larger AUC (*Hanley & Mcneil, 1982*) value by weight coefficient $\alpha$ (Eq. (6)) again.

$$\alpha = [auc(Z^I) \div auc(Z^B)]^\psi \tag{6}$$

where $auc()$ is a function used to calculate the value of AUC, $\psi$ is an amplification factor of the ratio $\frac{auc(Z^I)}{auc(Z^B)}$.

Especially, we should not use the method of reducing the element values of $Z^I$ or $Z^B$ to make their element values of the same order of magnitude, because it may result in invalid results due to the element values are too small. So, according to the values of coefficients $\alpha$ and $\lambda$, we divide the methods of linearly concatenating $Z^I$ and $Z^B$ into four cases as follows.

$$Z = (\overrightarrow{z_{v_0}}, \overrightarrow{z_{v_1}}, \ldots, \overrightarrow{z_{v_{(|V|-1)}}})^T = \begin{cases} [(\alpha * \lambda * Z^I) \| Z^B], & \text{if } \lambda \geq 1 \text{ and } \alpha \geq 1 \\ [(\lambda * Z^I) \| \dfrac{Z^B}{\alpha}], & \text{if } \lambda \geq 1 \text{ and } \alpha < 1 \\ [(\alpha * Z^I) \| \dfrac{Z^B}{\lambda}], & \text{if } \lambda < 1 \text{ and } \alpha \geq 1 \\ [(Z^I) \| \dfrac{Z^B}{\alpha * \lambda}], & \text{if } \lambda < 1 \text{ and } \alpha < 1 \end{cases} \tag{7}$$

where the operator $[\cdot \| \cdot]$ denotes concatenation, $\alpha$ is an enlarging coefficient to make the joint embedding matrix $Z$ more similar to $Z^I$ or $Z^B$ who has the higher AUC value, and $\lambda$ (Eq. (5)) denotes the enlargement factor who try to be adjusted to make the element values of $\lambda * Z^I$ and $Z^B$ or $Z^I$ and $\dfrac{Z^B}{\lambda}$ in the same order of magnitude.

For the case of $[(\alpha * \lambda * Z^I) \| Z^B]$ ($\lambda \geq 1$ and $\alpha \geq 1$) in Eq. (7), $Z$ is displayed in matrix form as follows:

$$Z = \begin{bmatrix} \alpha * \lambda * x_{00} & \cdots & \alpha * \lambda * x_{0(\tau-1)} & b_{00} & \cdots & b_{0(d-1)} \\ \alpha * \lambda * x_{10} & \cdots & \alpha * \lambda * x_{1(\tau-1)} & b_{10} & \cdots & b_{1(d-1)} \\ \vdots & \ddots & \vdots & \vdots & \ddots & \vdots \\ \alpha * \lambda * x_{(|V|-1)0} & \cdots & \alpha * \lambda * x_{(|V|-1)(\tau-1)} & b_{(|V|-1)0} & \cdots & b_{(|V|-1)(d-1)} \end{bmatrix},$$

where each row of $Z$ is the final joint embedding vector $\overrightarrow{z_v}$ for node $v$ based on the framework of IBE and $x_{ij}$ ($0 \leq i < |V|$, $0 \leq j < \tau$) is the element of $\overrightarrow{z_{v_i}^I}$, $b_{ij}$ ($0 \leq j < d$) is the element of $\overrightarrow{z_{v_i}^B}$. The other three cases of Eq. (7) have similar matrix representations.

## EXPERIMENTS

In this section, we describe our datasets, baseline models and present the experimental results to demonstrate the performance of the IBE framework in link prediction tasks. The source code and datasets can be obtained from https://github.com/swurise/IBE.

### Datasets

In Table 1, we consider the following real-world network datasets. BlogCatalog (http://networkrepository.com/soc-BlogCatalog.php) is asocial blog directory. The dataset contains 39 topic labels, 10,312 users, and 667,966 links. Zhihu is the largest online Q & A website in China. Users follow each other and answer questions on this site. We randomly crawl 10,000 active users from Zhihu and take the descriptions of their concerned topics as text information (Tu et al., 2017). The topics of Zhihu are obtained by the fastText model (Joulin et al., 2016) and the ODP of predefined topic categories (Haveliwala, 2002). The fastText presents the hierarchical softmax based on the Huffman tree to improve

**Table 1 Statistics of the real-world information networks.**

| Dataset Name | Social network | Language network | | Citation network | |
| --- | --- | --- | --- | --- | --- |
| | BlogCatalog | Zhihu | Wiki | Cora | Citeseer |
| Nodes | 10,312 | 10,000 | 2,408 | 2,277 | 3,312 |
| Edges | 667,966 | 43,894 | 17,981 | 5,214 | 4,732 |
| Attributes | – | 10,000 | – | 2,277 | – |
| Number of topics ($\tau$) | 39 | 13 | 17 | 7 | 6 |

the softmax classifier taking advantage of the fact that classification is unbalanced in CBOW (*Joulin et al., 2016*). WiKi contains 2,408 documents from 17 classes and 17,981 edges between them. Cora is a research paper classification citation network constructed by *McCallum et al. (2000)*. After filtering out papers without text information, 2,277 machine learning papers are divided into seven categories and 36 subcategories in this network. Citeseer is divided into six communities: Agents, AI, DB, IR, ML, and HCI and 4,732 edges between them. Similar to Cora, it records the citing and cited information between papers.

## Baselines

To validate the performance of our approach, we employ several state-of-the-art network embedding methods as baselines to compare with our IBE framework. A number of existing embedding methods are introduced as follows.

- CANE (*Tu et al., 2017*) learns context-aware embeddings with mutual attention mechanism for nodes, and the semantic relationship features are extracted between nodes. It jointly leverages network structure and textural information by regarding text content as a special kind of node.
- DeepWalk (*Perozzi, Al-Rfou & Skiena, 2014*) transforms a graph structure into a sample set of linear sequences consisting of nodes using uniform sampling. These linear sequences are transported to the Skip-Gram model to learn the distributed node embeddings.
- HOPE (*Ou et al., 2016*) is a graph embedding algorithm, which is scalable to preserve high-order proximities of large scale graphs and capable of capturing the asymmetric transitivity.
- LAP (*Belkin & Niyogi, 2001*) is a geometrically motivated algorithm for constructing a representation for data sampled from a low dimensional manifold embedded in a higher dimensional space.
- LINE (*Tang et al., 2015*) learns node embeddings in large-scale networks using first-order and second-order proximity between the nodes.
- Node2vec (*Grover & Leskovec, 2016*) has the same idea as DeepWalk using random walk sampling to get the combinational sequences of node context, and then the network embeddings of nodes are obtained by using the method of word2vec.

- GCN (*Kipf & Welling, 2016*) model uses an efficient layer-wise propagation rule based on a localized first-order approximation of spectral graph convolutions. The GCN model is capable of encoding graph structure and node features in a scalable approach for semi-supervised learning.
- GAT (*Veličković et al., 2018*) is a convolution-style neural network that operates on graph-structured data, leveraging masked self-attentional layers to address the shortcomings of methods based on graph convolutions. GAT enables implicitly specifying different weights to different nodes within a neighborhood.

## Evaluation metrics and parameter settings

We randomly divide all edges into two sets, training set and testing set, and take a standard evaluation metric AUC scores (area under the ROC curve) (*Hanley & Mcneil, 1982*) as evaluation metrics to measure the link prediction performance. AUC represents the probability that nodes in a random unobserved link are more similar than those in a random nonexistent link. Because the number of topics is different in each dataset, we use $\tau$ to denote the maximum number of topics for each dataset. In concatenating weight coefficient $\alpha = [auc(Z^I) \div auc(Z^B)]^\psi$, we set the factor $\psi$ equal to 4 except that $\psi$ has a specified value.

## Experimental results

For experiments, we evaluate the effectiveness and efficiency of our IBE on five networks for the link prediction task. For each dataset, we compare the AUC data of basic embedding matrix $Z^B$ generating by one of the existing embedding methods, the information behavior feature vectors $Z^I$, and their joint embedding vectors $Z$ generating by the framework IBE. We employ six state-of-the-art embedding methods as baselines, including Node2vec, DeepWalk, LINE, LAP, CANE, HOPE, for comparisons with their extending frameworks IBE in the following experiments.

Table 2 compares AUCs over five datasets. By concatenating $Z^B$ and $Z^I$ linearly, the joint embedding vectors $Z$ achieves the best performance. Especially on the BlogCatalog and Zhihu datasets, the AUC values of the joint embedding vectors $Z$ are higher, compared with their baselines, than 14.6% and 27.6% respectively on average. One of the reasons may be that the average AUC of information behavior feature vectors $Z^I$ are 81.7 and 82.4 respectively, which are more than 11% higher compared with the other three datasets on average. The other reason is that the maximum topic number $\tau$ of BlogCatalog and Zhihu are larger than those of Cora and Citeseer. In Table 2, we can also see that most of the AUC values of the joint embedding vectors $Z$ for datasets Wiki and Cora exceed 90%. The reason is that their AUC values of baselines are relatively high, most of them are more than 85%. On the Citeseer dataset of Table 2, we also can see that the improvement of the AUC values of the joint embedding vectors $Z$ is less, compared with their baselines. The result can be explained by the fact that the AUC values of information behavior feature vectors $Z^I$ are all low, about 55% and the direct reason for the low AUC values is that the number of topics is too small. Due to the number of topics is small, the topic

**Table 2 The AUC values of $Z^I$, $Z^B$ (*baseline*) and $Z$ (*baseline*) in different datasets where '(*baseline*)' in $Z^B$ (*baseline*), $Z$ (*baseline*) is to distinguish all kind of network embeddings $Z^B$ and their extension embeddings $Z$, and the $\psi$ equals 4 except that $\psi$ has a specified value.**

| Dataset Embedding | Social network | Language network | | Citation network | |
|---|---|---|---|---|---|
| | BlogCatalog ($\tau = 39$) | Zhihu ($\tau = 13$) | Wiki ($\tau = 17$) | Cora ($\tau = 7$) | Citeseer ($\tau = 6$) |
| $Z^I$ | 81.5 | 81.5 | 69.1 | 69.9 | 54.5 |
| $Z^B$ (*CANE*) | – | 71.2 | – | 94.5 | – |
| $Z$ (*CANE*) | – | 85.6 | – | 94.7 | – |
| $Z^B$ (*DeepWalk*) | 72.1 | 58.2 | 90.6 | 89.5 | 84.5 |
| $Z$ (*DeepWalk*) | 85.0 | 84.9 | 93.5 | 89.5 ($\psi = 1$) | 85.7 |
| $Z^B$ (*HOPE*) | 83.3 | 64.3 | 92.6 | 84.6 | 70.2 |
| $Z$ (*HOPE*) | 83.0 | 82.0 ($\psi = 6$) | 92.7 | 86.7 | 71.2 |
| $Z^B$ (*LAP*) | 75.2 | 74.2 | 92.2 | 87.9 | 80.3 |
| $Z$ (*LAP*) | 87.3 | 86.0 | 93.2 | 89.1 | 81.5 |
| $Z^B$ (*LINE*) | 59.1 | 51.6 | 87.8 | 78.3 | 69.8 |
| $Z$ (*LINE*) | 82.1 | 82.8 | 88.1 ($\psi = 7$) | 80.7 | **73.8** |
| $Z^B$ (*Node2vec*) | 71.0 | 56.1 | 89.0 | 86.7 | 83.2 |
| $Z$ (*Node2vec*) | 85.2 | 84.5 ($\psi = 6$) | 93.1 | 88.1 | 84.5 ($\psi = 2$) |
| $Z^B$ (*GCN*) | – | – | – | 90.2 | 89.7 |
| $Z$ (*GCN*) | – | – | – | 93.4 | 91.8 |
| $Z^B$ (*GAT*) | – | – | – | 92.6 | 90.2 |
| $Z$ (*GAT*) | – | – | – | 94.5 | 91.7 |

subdivision degree of nodes is low, and the classification of node labels will not be too detailed.

In general, in Table 2, it can be seen that the AUC values of concatenating embedding vectors $Z$ are higher than that of $Z^B$ and $Z^I$, which indicates that the concatenating method can properly integrate the features of all parties.

**Ablation experiments:** To investigate the effectiveness of TNE, we perform several ablation studies. In Table 2, the $Z^B$ is the basic embedding generated by an existing model. The $Z^I$ is obtained by the TNE model. A joint embedding $Z$ is obtained by adding a $Z^I$ to a $Z^B$, shown in Fig. 2 and Eq. (2). We observe that the quality of the joint embedding $Z$ is better than itself $Z^B$.

## Parameter sensitivity analysis

We further performed parameter sensitivity analysis in this section, and the results are summarized in Figs. 4 and 5. Due to space limitations, we only take the dataset of Wiki, Zhihu as an examples to estimate the topic number j ($0 \le j \le \tau$) and the amplification factor $\psi$ of vector concatenation can affect the link prediction results.

**The topic number j ($0 \le j \le \tau$):** In Fig. 4, we illustrate the relationship between the number of topics and link prediction, where the order of selection of node topics is random. When $j = 0$, $Z^I$ does not exist, $Z$ degenerates to $Z^B$. As shown in Fig. 4, we can see that as $j$ increases from 1 to $\tau$, $Z^B$ linearly combines with more topic-based feature dimensionality from $Z^I$, and the AUC values keep changing. When the AUC values of $Z^B$

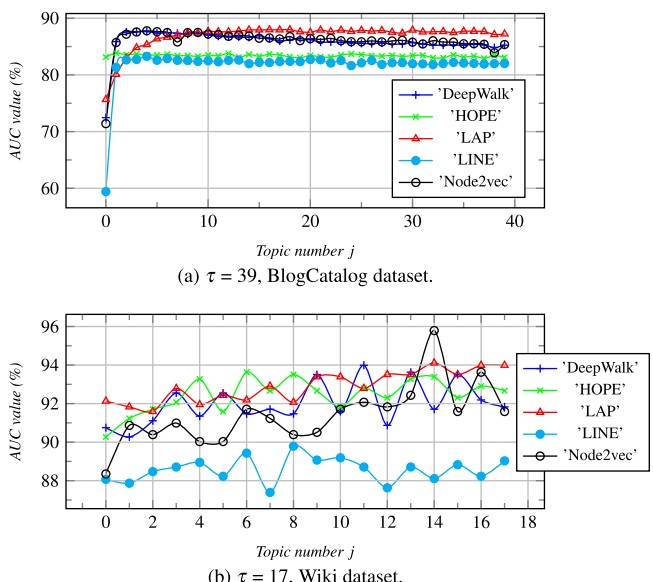

(a) $\tau = 39$, BlogCatalog dataset.

(b) $\tau = 17$, Wiki dataset.

**Figure 4** $\psi = 5$. With the increase of $\tau$, the AUC values are calcuiated for different datasets. $j$ is a topic number, if j = 0 then $Z = Z^B$ else $Z = [Z^B \| Z^I]$.

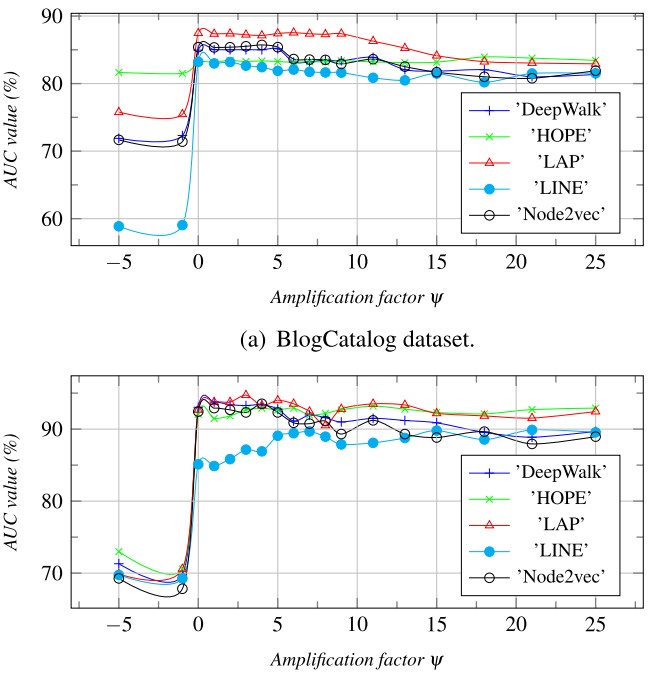

(a) BlogCatalog dataset.

(b) Wiki dataset.

**Figure 5** **The topic number is $\tau$, and $\psi$ is an amplification factor.** With the increase of $\psi$, the AUC values are calculated for different datasets.

are below 82% in Fig. 4A, AUC values increase sharply with an increase of $j$. When the AUC values of $Z^B$ are higher than a certain critical value, the AUC values increase more slowly or even stop growing with an increase of $j$.

So, we can see that when the number of topics is large in a dataset, each node can be classified in detail by the topic classification labels, which helps to improve the AUC values using a small number of topics. These also show that the AUC values of the concatenating embedding vectors $Z$s will be higher than that of all parties for concatenating, that is $Z^I$s and $Z^B$s, but it will not increase indefinitely.

**The amplification factor $\psi$ of vector concatenation:** $\psi$ is an amplification factor for $\alpha$ (Eq. (6)) which is a weight coefficient for enlarging the element values of $Z^I$ or $Z^B$ who has the larger AUC (Eq. (7)). From Fig. 5, we can see that the AUC value, when $\psi$ is less than 0, is less than that when $\psi$ is greater than 0. The reason is that the weight coefficient $\alpha$, when $\psi$ is less than 0, enlarges the $Z^I$ or the $Z^B$ who has the smaller AUC value. As a result, the joint embedding $Z$ is more similar to one that has a lower AUC value. When $\psi$ is between 1 and 5, the prediction result is the best. However, when the value of $\psi$ increases gradually, the AUC values decrease slightly and tend to the $Z^I$s or the $Z^B$s who have the larger AUC values.

## CONCLUSION AND FUTURE WORK

This article has presented an effective network embedding framework IBE, which can easily incorporate topology features, attribute features, and features of topic-based information behavior into network embedding. In IBE, we linearly combinate $Z^I$ and $Z^B$ to generate node joint embedding matrix $Z$. To get the $Z^I$, we have proposed the TNE model to extract the node's information behavior features. The model contains an ISR algorithm to generate the topic-sensitive embedding matrix ($\Gamma$) and a $\Gamma$ matrix transposing algorithm to transpose $\Gamma$ matrix into the information behavior feature matrix $Z^I$ for nodes eventually. Experimental results in various real-world networks have shown the efficiency and effectiveness of joint embedding vectors in link prediction. In the future, we plan to investigate other methods of extracting features that may better integrate with the TNE model. Moreover, we will further investigate how the TNE model works in heterogeneous information networks.

### Funding

This article is funded by the National Natural Science Foundation of China (62032019, 61732019, 61672435), and the Capacity Development Grant of Southwest University (SWU116007). The funders had no role in study design, data collection and analysis, decision to publish, or preparation of the manuscript.

### Grant Disclosures

The following grant information was disclosed by the authors:
National Natural Science Foundation of China: 62032019, 61732019, 61672435.
Capacity Development Grant of Southwest University: SWU116007.
## Competing Interests

Jun Pang is an Academic Editor for PeerJ.

## Author Contributions

- Ganglin Hu conceived and designed the experiments, performed the experiments, analyzed the data, performed the computation work, prepared figures and/or tables, authored or reviewed drafts of the article, and approved the final draft.
- Jun Pang analyzed the data, authored or reviewed drafts of the article, and approved the final draft.
- Xian Mo analyzed the data, authored or reviewed drafts of the article, contributed analysis tools, and approved the final draft.

## Data Availability

The source code and datasets are available at GitHub: https://github.com/swurise/IBE.

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
