# Peer review of "Effective attributed network embedding with information behavior extraction"

_PeerJ Computer Science, doi:10.7717/peerj-cs.1030_

## Round 0.1 · original submission · Major Revisions

A revision is needed before further processing. Please provide a detailed response letter. Please also note that I do not expect you to cite recommended references unless it is crucial. I look forward to having your revised version. Thanks.

·

Basic reporting

Throughout the manuscript, this paper uses clear and unambiguous professional English. Although adequate literature references and field background/context are provided, the literature discussion is missing a few related works. The proposed method is self-contained, with results correlating to hypotheses. The formal results should contain precise definitions of all terms and theorems, as well as detailed proofs.

Experimental design

Numerous evaluation results corroborate the proposed method. The research question is clearly defined, pertinent, and meaningful. It is stated how research fills a knowledge gap that has been identified. Extensive investigation conducted to the highest technical and ethical standards. Methods described in sufficient detail and detail to permit replication.

Validity of the findings

The underlying experimental results have been provided in its entirety; they are robust, statistically sound, and well-controlled. Conclusions are succinct, relevant to the original research question, and limited to supporting data.

Additional comments

This manuscript is written in clear and unambiguous professional English throughout. While adequate references to the literature and background/context of the field are provided, the literature discussion is missing a few related works [1,2,3,4,5]. The proposed method is self-contained and produces results that are consistent with hypotheses. All terms and theorems should have precise definitions in the formal results, as well as detailed proofs.

[1] WebFormer: The Web-page Transformer for Structure Information Extraction---WWW 2022
[2] A vector-based representation to enhance head pose estimation---WACV 2021
[3] Sg-net: Spatial granularity network for one-stage video instance segmentation---CVPR 2021
[4] DenserNet: Weakly supervised visual localization using multi-scale feature aggregation---AAAI 2021
[5] Video object detection for autonomous driving: Motion-aid feature calibration---Neurocomputing 2021

Reviewer 2 ·

Basic reporting

This manuscript captures the structural and attribute features of the network using existing network embedding algorithms, and proposes the TNE model to extract additional information behavior features. Finally, the link prediction performance of the network is improved.

1.The number of references in the past three years (2019-2022) is only two, and it is recommended to add more latest references.

Experimental design

1. In EXPERIMENTS, the latest comparison algorithm is from 2017, and it is recommended to add newer algorithms for comparison.
2.From the descriptions of Section 3 and Section 4, it can be seen that the extraction of node information behavior features requires additional topic information. From the experiment, the topic information seems to be the category information of the node. That is, in Table 2, the algorithm of the manuscript is supervised, while the algorithm used for comparison is unsupervised, that seems unfair. It is recommended to add some supervised/semi-supervised graph neural network, such as GAT, GIN, for comparison (the amount of information (category information) used should be consistent).
3.The overall performance of the algorithm is improved, but the increase in time complexity caused by adding information behavior features is worrying, such as the calculation of parameters \alpha. It is recommended to add time complexity analysis, including elaboration of the average number of cycles (convergence rate) of the ISR algorithm.
4.The number of topics in Zhihu described in Table 1 is empty, what is the basis for setting the number of topics as 13 in the experiment?
5.It is recommended to include a graph of the Importance Score distribution of nodes in the experiment to eliminate the concern that the importance may all be clustered into a very small number of nodes.
6.The manuscript adds Parameter sensitivity analysis, that is good. In the analysis of the topic number j, it is best to add a description of the implementation, for example, for nodes with unknown topics, how are they handled in the ISR algorithm?

Validity of the findings

no comment

Additional comments

1.The manuscript does not mention the algorithms incorporating information behavior in INTRODUCTION or RELATED WORK, which seems to imply that this manuscript is the first network embedding work that incorporates information behavior. Is this true? If so, please state clearly in ABSTRACT and INTROCUCTION. If not, please add the description of the corresponding algorithms, and add the comparative experiment of some corresponding algorithms in the experimental section.
2.Are the information behavior features and node categories the same? If so, a large number of related papers have proposed, that adding information behavior features cannot be a bright spot. If not, does the dataset in the experiment reflect the difference between them?

Reviewer 3 ·

Basic reporting

This paper presents an embedding method of attribute network through ISR algorithm and topic sensitive embedding matrix. Then, the above embedding is spliced into the embedding obtained by the existing characterization methods through appropriate methods. In this paper, experiments are conducted to verify the effectiveness of the method. However, this paper also has some issues.

1. The main purpose of this method is to obtain the embedding of attribute network, but it is not compared with the latest attribute network embedding methods. It is suggested to add relevant baseline algorithms, such as GCN[1], GAT[2] and AM-GCN[3].
2. There are lots of advanced attributed network embedding methods that should be discussed in the related work, such as MAGNN[4], FAME[5].
3. The datasets used in this paper are relatively small, and efficiency verification should be carried out on large-scale datasets.
4. Ablation experiments should be conducted to verify the effectiveness of model components.

[1] Thomas N. Kipf and Max Welling. 2017. Semi-Supervised Classification with Graph Convolutional Networks. In ICLR.
[2] Petar Velickovic, Guillem Cucurull, Arantxa Casanova, Adriana Romero, Pietro Liò, and Yoshua Bengio. 2018. Graph Attention Networks. In ICLR.
[3] Xiao Wang, Meiqi Zhu, Deyu Bo, Peng Cui, Chuan Shi, and Jian Pei. 2020. Am-gcn: Adaptive multi-channel graph convolutional networks. In KDD. 1243–1253.
[4] Xinyu Fu, Jiani Zhang, Ziqiao Meng, and Irwin King. 2020. Magnn: Metapath aggregated graph neural network for heterogeneous graph embedding. In WWW. 2331–2341.
[5] Zhijun Liu, Chao Huang, Yanwei Yu, Baode Fan, and Junyu Dong. 2020. Fast Attributed Multiplex Heterogeneous Network Embedding. In CIKM. 995–1004.

Experimental design

no comment

Validity of the findings

no comment

Additional comments

no comment

---

## Round 0.2 · accepted · Accept

The paper can be accepted. Congratulations.

Reviewer 3 ·

Basic reporting

This is a revised manuscript and the authors have revised most of the comments raised by the reviewers. The quality of manuscript has been signficantly improved.

Experimental design

Experimental evaluation has been improved.

Validity of the findings

The proposed TNE model improves the performance of existing embedding models.

Additional comments

NA